# Detection of Interaction Effects in a Nonparametric Concurrent Regression Model

**DOI:** 10.3390/e25091327

**Published:** 2023-09-12

**Authors:** Rui Pan, Zhanfeng Wang, Yaohua Wu

**Affiliations:** 1School of Data Science, University of Science and Technology of China, Hefei 230026, China; kevinpan@mail.ustc.edu.cn; 2Department of Statistics and Finance, Management School, University of Science and Technology of China, Hefei 230026, China; wyh@ustc.edu.cn

**Keywords:** model selection, *L*_1_ criterion, reproducing kernel Hilbert space, smoothing spline

## Abstract

Many methods have been developed to study nonparametric function-on-function regression models. Nevertheless, there is a lack of model selection approach to the regression function as a functional function with functional covariate inputs. To study interaction effects among these functional covariates, in this article, we first construct a tensor product space of reproducing kernel Hilbert spaces and build an analysis of variance (ANOVA) decomposition of the tensor product space. We then use a model selection method with the $L_{1}$ criterion to estimate the functional function with functional covariate inputs and detect interaction effects among the functional covariates. The proposed method is evaluated using simulations and stroke rehabilitation data.

## 1. Introduction

Functional data can be found in various fields, such as biology, economics, engineering, geology, medicine and psychology. Recently, statistical methods and theories on functional data were widely studied ([1,2,3,4,5,6]). Functional data sometimes have more complicated structures. For example, the motivation data of this paper, stroke rehabilitation data, utilized a collection of 3D video games known as Circus Challenge to enhance upper limb function in stroke patients ([7,8,9]). The patients were scheduled to play the movement game over three months at specified times. At each visit time *t*, the level of impairment of stroke subject *i* was measured using the CAHAI (Chedoke Arm and Hand Activity Inventory) score, denoted as $y_{i}\left(t\right)$, and movements of upper limbs of patients, such as forward circle movement and sawing movement, were also recorded. The movement data at time *t* and frequency *s* from the *i*th patient were denoted as $x_{i}\left(t,s\right)$. Determining a way to model the relationship of $y_{i}\left(t\right)$ and functional $x_{i}\left(t,\;\cdot \;\right)$ is key to studying whether the movements are helpful to the rehabilitation level of the stroke patient or not. Furthermore, there is a question of whether there are interaction effects among the movements on the stroke patient’s rehabilitation. Zhai et al. [9] developed a nonparametric concurrent regression model to study the relationship between functional movements and the CAHAI score. However, they did not consider the interaction effects of functional movements on the CAHAI score. We aim to examine the interaction effects of movements on CAHAI scores and predict the rehabilitation level of stroke patients in this paper.

In this paper, we apply the following nonparametric concurrent regression model (NCRM) to model stroke rehabilitation data: (1)$$\begin{array}{c} y_{i}\left(t\right)=f\left(t,x_{i}\left(t,\;\cdot \;\right)\right)+\epsilon_{i}\left(t\right),\;\;i=1,\ldots ,n, \end{array}$$
where *f* is a bivariate functional to be estimated nonparametrically, response $y_{i}\left(t\right)$ is a function of *t*, covariate $x_{i}\left(t,\;\cdot \;\right)$ is a vector of functional with length *q*, and $\epsilon_{i}\left(t\right)$ is a random error. To explore the interaction effects among components of covariates $x_{i}\left(t,\;\cdot \;\right)$, we use the smooth spline analysis of variance (SS ANOVA) method [10,11] to decompose regression function *f*.

A multivariate function can be decomposed of main effects and interaction effects via the SS ANOVA method ([10,11]). When the dimension of covariates *q* is large, the decomposed model contains a large number of interaction effects. Even if only the main effects and second-order interaction terms are investigated, the order of the number of decomposition terms is $O(q^{2})$, which leads to a highly complicated model. To model stroke rehabilitation data with q=3, there are 22 terms, including the main effects and interaction effects. This challenges the estimation method for the NCRM model. To avoid this shortcoming, Zhai et al. [9] took all functional covariates as a whole and did not consider interaction effects among covariates. Following [12], this paper conducts a model selection method for the NCRM model with all main effects and interaction effects. In this method, the regression function is estimated and significant components of the decomposition are selected simultaneously.

Model selection is a crucial step in building statistical models that accurately capture the relationships between variables ([13,14]). It can choose the most suitable model from a set of candidate models based on certain criteria such as goodness-of-fit, predictive performance, and interpretability. Based on the SS ANOVA approach, model selection is crucial to determine the contribution of each component of the decomposition to the overall variance of the response variable. Several methods have been proposed for selecting models with SS ANOVA, including forward selection, backward elimination, and stepwise regression ([15,16,17,18,19,20]). However, these methods are limited in their ability to handle high-dimensional data and identify complex interactions among variables. Hence, regularization methods such as the $L_{1}$ penalty have gained popularity in recent years ([12,21,22,23,24,25]), which allow for the selection of sparse and robust models. For example, Zhang et al. [23] developed a nonparametric penalized likelihood method with the likelihood basis pursuit and used it for variable selection and model construction. Lin and Zhang [22] proposed a component selection and smoothing method for multivariate nonparametric regression models by penalizing the sum of component norms of SS ANOVA. Furthermore, Wang et al. [12] developed a unified framework for estimation and model selection methods in nonparametric function-on-function regression, which performs well when using $L_{1}$ penalty methods for model selection. Dong and Wang [24] proposed a nonparametric method for learning conditional dependence structure in graph models by applying $L_{1}$ regularization to detect the neighborhoods of edges, where SS ANOVA decomposition is used to depict interaction effects of edges in the graph model. In this paper, we borrowed the $L_{1}$ regularization idea to build model selection by penalizing the sum of norms of the ANOVA decomposition components for the NCRM model. In addition, Bayesian analysis methods can also be used to study interaction effects; for example, Ren et al. [26] proposed a novel semiparametric Bayesian variable selection model for investigating linear and nonlinear gene–environment interactions simultaneously, allowing for structural identification.

This paper proposes an estimation and model selection approach for the NCRM model (Equation 1). Following [12,22], the SS ANOVA decomposition for the tensor product space of the reproducing kernel Hilbert spaces (RKHS) is constructed, and the $L_{1}$ penalty approach for the components of the decomposition is implemented. We use estimation procedures under either an $L_{1}$ or a joint $L_{1}$ and $L_{2}$ penalty to fit teh NCRM model. We study the interaction effects of the covariate $x_{i}\left(t,\;\cdot \;\right)$ in model (Equation 1) via ANOVA decomposition of the regression function, where the tensor product RKHS is built based on Gaussian kernels. The decomposition is different from that of Zhai et al. [9], where they took the covariate as a whole variable and did not consider their interaction effects. Based on the decomposition, model selection with the tensor product RKHS is conducted using the $L_{1}$ penalty method. With regards to the covariate $x_{i}\left(t,\;\cdot \;\right)$, the models from Wang et al. [12] are not suitable to analyze the stoke data. In this paper, we apply the proposed method to stroke rehabilitation data and study the relationship of the movements and the patient’s CAHAI score. Besides the main effects, the interaction effect of the movements is also detected.

The remainder of the article is organized as follows. In Section 2, we present the tensor product RKHS with the Gaussian kernel and the SS ANOVA decomposition of the regression function. In Section 3, we show model selection and estimation procedures. The simulation study and application of stroke rehabilitation data are presented in Section 4 and Section 5. We conclude in Section 6.

## 2. Nonparametric Concurrent Regression Model

For the NCRM model (Equation 1), we consider $x_{i}\left(t,\;\cdot \;\right)=\left(x_{i1}\left(t,\;\cdot \;\right),\ldots ,x_{iq}\left(t,\;\cdot \;\right)\right)$, where $x_{ij}\left(t,s\right):S\to R$ for any fixed time t∈T is a function of *s* within a space denoted by $X_{j}$, j∈1,⋯,q. Generally, *t* and *s* can be transformed into [0,1]. For simplicity, we let T=[0,1] and S=[0,1] and let $X_{j}\subset L^{2}\left[0,1\right]$, j=1,⋯,q, which are independent of *t*. Furthermore, we assume that $y_{i}\left(t\right)\in Y\subset L^{2}\left[0,1\right]$ and $\epsilon_{i}\left(t\right)$ for i=1,…,n are identically and independently distributed in $L^{2}\left[0,1\right]$ with mean zero and $\int_{0}^{1}E\left[\epsilon_{i}{\left(t\right)}^{2}\right]dt<\infty$. It is shown that the regression function *f* is a functional function with an independent covariate $x_{i}\left(t,\;\cdot \;\right)$. To provide a nonparametric estimation of *f*, the SS ANOVA decomposition method is used to construct a tensor product space of RKHS to which *f* belongs.

When *f* is treated as a function with respect to the first augment t∈T, we consider the Sobolev space [10],
(2)$$H^{\left(1\right)}=\left\{f:\;f\;\mathrm{and}\;f^{\prime }\;\mathrm{absolutely}\;\mathrm{continuous},\;\int_{0}^{1}{\left(f^{\prime \prime }\right)}^{2}dt<\infty \right\},$$
where $H^{\left(1\right)}$ can be rewritten as
$$H^{\left(1\right)}=\left\{1\right\}⊕\left\{t\right\}⊕H_{2}^{\left(1\right)},$$
where {1} is a constant space, {t} is a linear function space with *t* as an independent variable. $H_{2}^{\left(1\right)}$ is a smooth function space orthogonal to the constant space and the linear function space. Reproducing kernels (RK) for these three subspaces are $K_{0}^{\left(1\right)}\left(t,t^{\prime }\right)=1$, $K_{1}^{\left(1\right)}\left(t,t^{\prime }\right)=k_{1}\left(t\right)k_{1}\left(t^{\prime }\right)$, and $K_{2}^{\left(1\right)}\left(t,t^{\prime }\right)=k_{2}\left(t\right)k_{2}\left(t^{\prime }\right)-k_{4}(|t-t^{\prime }\left|\right)$, where $k_{1}$, $k_{2}$ and $k_{4}$ are defined as
$$\begin{array}{cc} k_{1}\left(x\right) & =x-0.5,\\ k_{2}\left(x\right) & =\frac{1}{2}\left\{k_{1}^{2}\left(x\right)-\frac{1}{12}\right\},\\ k_{4}\left(x\right) & =\frac{1}{24}\left\{k_{1}^{4}\left(x\right)-\frac{1}{2}k_{1}^{2}\left(x\right)+\frac{7}{240}\right\}. \end{array}$$

For functional augments x(t,·), RK and its corresponding RKHS for *f* as a function of functions in $X=X_{1}\times \cdots \times X_{q}$ are constructed as follows. For any $u_{j},u_{j}^{\prime }\in X_{j}$, we construct a Gaussian kernel as
(3)$$\begin{array}{c} K_{2,j}^{\left(2\right)}\left(u_{j},u_{j}^{\prime }\right)=\exp\left\{-\frac{{∥u_{j}-u_{j}^{\prime }∥}^{2}}{2}\right\}, \end{array}$$
where ${∥u_{j}∥}^{2}=\int_{0}^{1}u_{j}^{2}\left(s\right)ds$. We can show that when the space $X_{j}$ is a complete space, $K_{2,j}^{\left(2\right)}$ is a symmetric and strictly positive definite. The unique RKHS $H_{2,j}^{\left(2\right)}$ derived from $K_{2,j}^{\left(2\right)}$ is separable and does not contain any non-zero constants. To construct an SS ANOVA decomposition, we let $H_{j}^{\left(2\right)}=\left\{1\right\}⊕H_{2,j}^{\left(2\right)}$. Then, the tensor product space in this paper is $H^{\left(2\right)}=H_{1}^{\left(2\right)}⊗\cdots ⊗H_{q}^{\left(2\right)}$ with the following decomposition:(4)$$\begin{array}{cc} H^{\left(2\right)}= & H_{1}^{\left(2\right)}⊗\cdots ⊗H_{q}^{\left(2\right)}\\ = & \left\{1\right\}⊕H_{2,1}^{\left(2\right)}⊕\cdots ⊕H_{2,q}^{\left(2\right)}⊕\left\{H_{2,1}^{\left(2\right)}⊗H_{2,2}^{\left(2\right)}\right\}⊕\cdots ⊕\\ \; & \left\{H_{2,q-1}^{\left(2\right)}⊗H_{2,1}^{\left(2\right)}\right\}⊕\cdots ⊕\left\{H_{2,1-1}^{\left(2\right)}⊗\cdots ⊗H_{2,q}^{\left(2\right)}\right\}. \end{array}$$
Decomposition (Equation 4) is different from that of Zhai et al. [9] where $H^{\left(2\right)}$ is decomposed of constant space {1} and another RKHS not considering interaction among x(t,·).

Next, we consider the tensor product space $H=H^{\left(1\right)}⊗H^{\left(2\right)}$ which has the following decomposition:$$\begin{array}{cc} H= & \left(\left\{1\right\}⊕\left\{t\right\}⊕H_{2}^{\left(1\right)}\right)⊗\left(\left\{1\right\}⊕\left\{{⊕}_{j=1}^{q}H_{2,j}^{\left(2\right)}\right\}⊕\cdots ⊕\left\{H_{2,1}^{\left(2\right)}⊗\cdots ⊗H_{2,q}^{\left(2\right)}\right\}\right)\\ = & \left\{1\right\}⊕\left\{t\right\}⊕H_{2}^{\left(1\right)}⊕\left\{{⊕}_{j=1}^{q}H_{2,j}^{\left(2\right)}\right\}⊕\left\{{⊕}_{j=1}^{q}\left\{H_{2,j}^{\left(2\right)}\times \left\{t\right\}\right\}\right\}⊕\\ \; & \left\{{⊕}_{j=1}^{q}\left\{H_{2,j}^{\left(2\right)}\times H_{2}^{\left(1\right)}\right\}\right\}⊕\cdots ⊕\left\{H_{2,1}^{\left(2\right)}⊗\cdots ⊗H_{2,q}^{\left(2\right)}\right\}⊕\\ \; & \left\{H_{2,1}^{\left(2\right)}⊗\cdots ⊗H_{2,q}^{\left(2\right)}\times \left\{t\right\}\right\}⊕\left\{H_{2,1}^{\left(2\right)}⊗\cdots ⊗H_{2,q}^{\left(2\right)}⊗H_{2}^{\left(1\right)}\right\}. \end{array}$$
There, the null space {1}⊕{t} stands for the main effect of the parametric form of *t*, $H_{2}^{\left(1\right)}$ is the main effect of the non-parametric form of *t*, $H_{2,j}^{\left(2\right)}$ is the main effect of the non-parametric form of $u_{j}$, $\left\{t\right\}⊗H_{2,j}^{\left(2\right)}$ is the linear nonparametric interaction between *t* and $u_{j}$, $H_{2}^{\left(1\right)}⊗H_{2,j}^{\left(2\right)}$ is the nonparametric nonparametric interaction between *t* and $u_{j}$, and so on, $H_{2}^{\left(1\right)}⊗H_{2,1}^{\left(2\right)}⊗\cdots ⊗H_{2,q}^{\left(2\right)}$ is the nonparametric nonparametric interaction between *t* and *u*, where $u=(u_{1},\ldots ,u_{q})$. We denote $\phi_{1}\left(t,u\right)=1$ and $\phi_{2}\left(t,u\right)=k_{1}\left(t\right)$ as the basis functions of $H_{0}$. For example, with q=3, the RKs corresponding to the above sub-RKHS are
$$\begin{array}{cc} H_{0}:=\left\{1\right\}⊕\left\{t\right\}\;\; & ⟷\;\;K_{0}\left(\left(t,u\right),\left(t^{\prime },u^{\prime }\right)\right)=1+k_{1}\left(t\right)k_{1}\left(t^{\prime }\right),\\ H_{1}:=H_{2}^{\left(1\right)}\;\; & ⟷\;\;K_{1}\left(\left(t,u\right),\left(t^{\prime },u^{\prime }\right)\right)=K_{2}^{\left(1\right)}\left(t,t^{\prime }\right),\\ H_{1+j}:=H_{2,j}^{\left(2\right)}\;\; & ⟷\;\;K_{1+j}\left(\left(t,u\right),\left(t^{\prime },u^{\prime }\right)\right)=K_{2,j}^{\left(2\right)}\left(u_{j},u_{j}^{\prime }\right),\\ H_{4+j}:=\left\{t\right\}⊗H_{2,j}^{\left(2\right)}\;\; & ⟷\;\;K_{4+j}\left(\left(t,u\right),\left(t^{\prime },u^{\prime }\right)\right)=k_{1}\left(t\right)k_{1}\left(t^{\prime }\right)K_{2,j}^{\left(2\right)}\left(u_{j},u_{j}^{\prime }\right),\\ H_{7+j}:=H_{2}^{\left(1\right)}⊗H_{2,j}^{\left(2\right)}\;\; & ⟷\;\;K_{7+j}\left(\left(t,u\right),\left(t^{\prime },u^{\prime }\right)\right)=K_{2}^{\left(1\right)}\left(t,t^{\prime }\right)K_{2,j}^{\left(2\right)}\left(u_{j},u_{j}^{\prime }\right),\\ H_{8+j+l}:=H_{2,j}^{\left(2\right)}⊗H_{2,l}^{\left(2\right)}\;\; & ⟷\;\;K_{8+j+l}:=\left(\left(t,u\right),\left(t^{\prime },u^{\prime }\right)\right)=K_{2,j}^{\left(2\right)}\left(u_{j},u_{j}^{\prime }\right)K_{2,l}^{\left(2\right)}\left(u_{l},u_{l}^{\prime }\right),\\ H_{11+j+l}:=\left\{t\right\}⊗H_{2,j}^{\left(2\right)}⊗H_{2,l}^{\left(2\right)}\;\; & ⟷\;\;K_{11+j+l}\left(\left(t,u\right),\left(t^{\prime },u^{\prime }\right)\right)=k_{1}\left(t\right)k_{1}\left(t^{\prime }\right)K_{2,j}^{\left(2\right)}\left(u_{j},u_{j}^{\prime }\right)K_{2,l}^{\left(2\right)}\left(u_{l},u_{l}^{\prime }\right),\\ H_{14+j+l}:=H_{2}^{\left(1\right)}⊗H_{2,j}^{\left(2\right)}⊗H_{2,l}^{\left(2\right)}\;\; & ⟷\;\;K_{14+j+l}\left(\left(t,u\right),\left(t^{\prime },u^{\prime }\right)\right)=K_{2}^{\left(1\right)}\left(t,t^{\prime }\right)K_{2,j}^{\left(2\right)}\left(u_{j},u_{j}^{\prime }\right)K_{2,l}^{\left(2\right)}\left(u_{l},u_{l}^{\prime }\right),\\ H_{20}:=H_{2,1}^{\left(2\right)}⊗H_{2,3}^{\left(2\right)}⊗H_{2,3}^{\left(2\right)}\;\; & ⟷\;\;K_{20}\left(\left(t,u\right),\left(t^{\prime },u^{\prime }\right)\right)=\prod_{j=1}^{3}K_{2,j}^{\left(2\right)}\left(u_{j},u_{j}^{\prime }\right),\\ H_{21}:=\left\{t\right\}⊗H_{2,1}^{\left(2\right)}⊗H_{2,3}^{\left(2\right)}⊗H_{2,3}^{\left(2\right)}\;\; & ⟷\;\;K_{21}\left(\left(t,u\right),\left(t^{\prime },u^{\prime }\right)\right)=k_{1}\left(t\right)k_{1}\left(t^{\prime }\right)\prod_{j=1}^{3}K_{2,j}^{\left(2\right)}\left(u_{j},u_{j}^{\prime }\right),\\ H_{22}:=H_{2}^{\left(1\right)}⊗H_{2,1}^{\left(2\right)}⊗H_{2,3}^{\left(2\right)}⊗H_{2,3}^{\left(2\right)}\;\; & ⟷\;\;K_{22}\left(\left(t,u\right),\left(t^{\prime },u^{\prime }\right)\right)=K_{2}^{\left(1\right)}\left(t,t^{\prime }\right)\prod_{j=1}^{3}K_{2,j}^{\left(2\right)}\left(u_{j},u_{j}^{\prime }\right), \end{array}$$
for j,l=1,2,3 and j<l, where the left and right parts stand for the tensor product spaces and their corresponding RKs, respectively.

## 3. Model Selection and Estimation

We let the projection of *f* onto $H_{0}$ be $\sum_{k=1}^{2}d_{k}\phi_{k}\left(t,u\right)$, $u=(u_{1},\ldots ,u_{q})$, and $\left\{H_{1},\cdots ,H_{Q}\right\}$ be the sub-RKHS generated by the tensor product method in Section 2, where *Q* is a number of sub-RKHS. $L_{1}$ penalties are applied to coefficients $d_{k}$ for the space $H_{0}$ and components of the decomposition of *f* (projections of *f* onto $H_{j},j=1,\cdots ,Q$). We estimate *f* by minimizing the following penalized least squares: (5)$$\begin{array}{c} \frac{1}{n}\sum_{i=1}^{n}\left\{\int_{0}^{1}{\left(y_{i}\left(t\right)-f\left(t,x_{i}\left(t,\;\cdot \;\right)\right)\right)}^{2}dt\right\}+\lambda_{1}\sum_{k=1}^{2}w_{1k}\left|d_{k}\right|+\lambda_{2}\sum_{v=1}^{Q}w_{2,v}{∥P_{v}f∥}_{H}, \end{array}$$
where f∈H, $P_{v}$ is the projection operator onto $H_{j}$, ${∥\;\cdot \;∥}_{H}$ is a norm induced from H, $\lambda_{1}$ and $\lambda_{2}$ are tuning parameters, and $0\leq w_{1k},w_{2,v}<\infty$ are pre-specified weights. We may set $w_{11}=0$ when $\phi_{1}=1$ to avoid penalty to the constant function.

Since the response function is a stochastic process in the $L^{2}\left[0,1\right]$ space, there exists a set of orthogonal basis functions $\left\{\eta_{k}\left(t\right),k=1,2,\ldots \right\}$ in $L^{2}\left[0,1\right]$, where $\left\{\eta_{k}\left(t\right),k=1,2,\ldots ,n\right\}$ is an empirical functional principal component (EFPC) of $\left\{y_{1}\left(t\right),\cdots ,y_{n}\left(t\right)\right\}$ ([27]). We let $\nu_{ik}=<y_{i}\left(t\right),\eta_{k}\left(t\right)>$ and $L_{ik}f=\int_{0}^{1}f\left(t,x_{i}\left(t,\;\cdot \;\right)\right)\eta_{k}\left(t\right)dt$ for i=1,2,…,n and k=1,…,n. We assume that $\left\{L_{ik}\right\}$ are bounded linear functionals. With EFPC, functional data can be transformed to scalar data such that modeling and analysis can be conducted by using traditional statistical methods. It can show that the PLS (Equation 5) based on functional data $y_{i}\left(t\right)$ reduces to the following PLS based on scalar data $\left\{\nu_{ik}\right\}$:(6)$$\begin{array}{c} \frac{1}{n}\sum_{i=1}^{n}\sum_{k=1}^{n}\left\{{\left(\nu_{ik}-L_{ik}f\right)}^{2}\right\}+\lambda_{1}\sum_{k=1}^{2}w_{1k}\left|d_{k}\right|+\lambda_{2}\sum_{v=1}^{Q}w_{2,v}{∥P_{v}f∥}_{H}. \end{array}$$
By Lemma 3.1 in Wang et al. [12], minimizing the PLS (Equation 6) is equivalent to minimizing the following PLS:(7)$$\begin{array}{c} \frac{1}{n}\sum_{i=1}^{n}\sum_{k=1}^{n}\left\{{\left(\nu_{ik}-L_{ik}f\right)}^{2}\right\}+\lambda_{1}\sum_{k=1}^{2}w_{1k}\left|d_{k}\right|+\tau_{0}\sum_{v=1}^{Q}w_{2,v}\theta_{v}^{-1}{∥P_{v}f∥}_{H}^{2}+\tau_{1}\sum_{v=1}^{Q}w_{2,v}\theta_{v}, \end{array}$$
subject to $\theta_{v}\geq 0$ for 1≤v≤Q, where $\lambda_{1}$, $\tau_{0}$, $\tau_{1}$ are tuning parameters.

We let $H^{*}=H_{1}⊕\cdots ⊕H_{Q}$. To provide an RK with linear combination of its subspaces RK for the space of $H^{*}$, we define a new inner product in $H^{*}$,
(8)$$\begin{array}{c} <f,g{>}_{*}=\sum_{v=1}^{Q}w_{2,v}\theta_{v}^{-1}<P_{v}f,P_{v}g>, \end{array}$$
where <·,·> is the inner product in H. Under the new inner product, the RK of $H_{1}^{*}$ is
$$\begin{array}{c} K^{*}\left(\left(t,u\right),\left(t^{\prime },u^{\prime }\right)\right)=\sum_{v=1}^{Q}w_{2,v}^{-1}\theta_{v}K_{v}\left(\left(t,u\right),\left(t^{\prime },u^{\prime }\right)\right), \end{array}$$
where coefficient $\theta_{v}$ can measure the contribution of the components of the decomposition to the model. Next, we use the reproducing property of the kernel function to transform the infinite-dimensional optimization problem (Equation 7) into a finite-dimensional solution problem. We let $H_{1n}=\mathrm{span}\left\{\int_{0}^{1}K^{*}\left(\left(t,x\left(t,\;\cdot \;\right)\right),\left(t^{\prime },x_{i}\left(t^{\prime },\;\cdot \;\right)\right)\right)\eta_{k}\left(t^{\prime }\right)dt^{\prime },\;i=1,2,\ldots ,n,\;k=1,2,\ldots ,n\right\}$, which is a subspace of $H^{*}$. Then, any $f\in H^{*}$ can be decomposed as
$$\begin{array}{c} f=f_{0}+f_{1n}+\rho , \end{array}$$
where $f_{0}\in H_{0}$, $f_{1n}\in H_{1n}$, and $\rho \in H^{*}⊖H_{1n}$. We denote
$$K_{\left(t^{\prime },x_{i}\left(t^{\prime },\;\cdot \;\right)\right)}^{*}\left(t,x\left(t,\;\cdot \;\right)\right)=K^{*}\left(\left(t^{\prime },x_{i}\left(t^{\prime },\;\cdot \;\right)\right),\left(t,x\left(t,\;\cdot \;\right)\right)\right)$$
as the evaluation function of point $\left(t^{\prime },x_{i}\left(t^{\prime },\;\cdot \;\right)\right)$, and $f_{1}=f_{1n}+\rho$. Then, we can rewrite the PLS (Equation 7) as
(9)$$\begin{array}{cc} \; & \frac{1}{n}\sum_{i=1}^{n}\sum_{k=1}^{n}{\left\{\nu_{ik}-u_{ik}-<f_{1}\left(t^{\prime },x_{i}\left(t^{\prime },\;\cdot \;\right)\right),\eta_{k}\left(t^{\prime }\right)>\right\}}^{2}+\tau_{0}\sum_{v=1}^{Q}w_{2,v}\theta_{v}^{-1}{∥P_{v}f∥}_{H^{*}}^{2}+\lambda_{1}\sum_{k=1}^{2}w_{1k}\left|d_{k}\right|+\tau_{1}\sum_{v=1}^{Q}w_{2,v}\theta_{v}\\ = & \frac{1}{n}\sum_{i=1}^{n}\sum_{k=1}^{n}{\left\{\nu_{ik}-u_{ik}-<<f_{1},K_{\left(t^{\prime },x_{i}\left(t^{\prime },\;\cdot \;\right)\right)}^{*}{>}_{H^{*}},\eta_{k}\left(t^{\prime }\right)>\right\}}^{2}+\tau_{0}\sum_{v=1}^{Q}w_{2,v}\theta_{v}^{-1}{∥P_{v}f∥}_{H^{*}}^{2}+\lambda_{1}\sum_{k=1}^{2}w_{1k}\left|d_{k}\right|+\tau_{1}\sum_{v=1}^{Q}w_{2,v}\theta_{v}\\ = & \frac{1}{n}\sum_{i=1}^{n}\sum_{k=1}^{n}{\left\{\nu_{ik}-u_{ik}-<f_{1},\int_{0}^{1}K_{\left(t^{\prime },x_{i}\left(t^{\prime },\;\cdot \;\right)\right)}^{*}\eta_{k}\left(t^{\prime }\right)dt^{\prime }{>}_{H^{*}}\right\}}^{2}+\tau_{0}\sum_{v=1}^{Q}w_{2,v}\theta_{v}^{-1}{∥P_{v}f∥}_{H^{*}}^{2}+\lambda_{1}\sum_{k=1}^{2}w_{1k}\left|d_{k}\right|+\tau_{1}\sum_{v=1}^{Q}w_{2,v}\theta_{v}\\ = & \frac{1}{n}\sum_{i=1}^{n}\sum_{k=1}^{n}{\left\{\nu_{ik}-u_{ik}-<f_{1n},\int_{0}^{1}K_{\left(t^{\prime },x_{i}\left(t^{\prime },\;\cdot \;\right)\right)}^{*}\eta_{k}\left(t^{\prime }\right)dt^{\prime }{>}_{H^{*}}\right\}}^{2}+\\ & \tau_{0}\sum_{v=1}^{Q}w_{2,v}\theta_{v}^{-1}{∥P_{v}f_{1n}∥}_{H^{*}}^{2}+\tau_{0}\sum_{v=1}^{Q}w_{2,v}\theta_{v}^{-1}{∥P_{v}\rho ∥}_{H^{*}}^{2}+\lambda_{1}\sum_{k=1}^{2}w_{1k}\left|d_{k}\right|+\tau_{1}\sum_{v=1}^{Q}w_{2,v}\theta_{v}, \end{array}$$
where $u_{ik}=\int_{0}^{1}f_{0}\left(t^{\prime },x_{i}\left(t^{\prime },\;\cdot \;\right)\right)\eta_{k}\left(t^{\prime }\right)dt^{\prime }$. The first equality uses the reproducing property, and the third equality uses the fact that ρ is orthogonal to $H_{1n}$. Minimizing (Equation 9) must have ρ=0, and we obtain the following representer theorem:

**Theorem** **1**(Representer Theorem). *The solution to PLS (Equation 9) is*
(10)$$\begin{array}{c} f\left(t,x\left(t,\;\cdot \;\right)\right)=\sum_{j=1}^{2}d_{j}\varphi_{j}\left(t\right)+\sum_{v=1}^{Q}w_{2,v}^{-1}\theta_{v}\sum_{i=1}^{n}\sum_{k=1}^{n}c_{ik}\xi_{ik}\left(t,x\left(t,\;\cdot \;\right)\right), \end{array}$$
*where $\varphi_{1}\left(t\right)=1$, $\varphi_{2}\left(t\right)=k_{1}\left(t\right)$, and $\xi_{ik}\left(t,x\left(t,\;\cdot \;\right)\right)=\int_{0}^{1}K_{v}\left(\left(t,x\left(t,\;\cdot \;\right)\right),\left(t^{\prime },x_{i}\left(t^{\prime },\;\cdot \;\right)\right)\right)\eta_{k}\left(t^{\prime }\right)dt^{\prime }$.*

From this representer theorem, the PLS (Equation 9) reduces to
(11)$$\begin{array}{cc} \; & \frac{1}{n}\sum_{i=1}^{n}\sum_{k=1}^{n}{\left(\nu_{ik}-\sum_{j=1}^{2}a_{ikj}d_{j}-\sum_{v=1}^{Q}w_{2,v}^{-1}\theta_{v}\sum_{j=1}^{n}\sum_{l=1}^{n}c_{jl}b_{ikjl}\right)}^{2}\\ + & \lambda_{1}\sum_{k=1}^{2}w_{1k}\left|d_{k}\right|+\tau_{0}\sum_{v=1}^{Q}w_{2,v}^{-1}\theta_{v}\sum_{i=1}^{n}\sum_{k=1}^{n}\sum_{j=1}^{n}\sum_{l=1}^{n}c_{ik}b_{ikjl}c_{jl}+\tau_{1}\sum_{v=1}^{Q}w_{2,v}\theta_{v}, \end{array}$$
where $a_{ikj}=\int_{0}^{1}\varphi_{j}\left(t\right)\eta_{k}\left(t\right)dt$, $b_{ikjl}=\sum_{v=1}^{Q}w_{2,v}^{-1}\theta_{v}b_{ikjl}^{v}$, $b_{ikjl}^{v}=\int_{0}^{1}\xi_{jl}\left(t,x_{i}\left(t,\;\cdot \;\right)\right)\eta_{k}\left(t\right)dt$. We let $\Sigma =\sum_{v=1}^{Q}w_{2,v}^{-1}\theta_{v}\Sigma_{v}$, the (i+(k−1)n,j+(l−1)n)th element of $\Sigma_{v}$ is $b_{ikjl}^{v}$. We let $Y_{k}={\left(\nu_{1k},\ldots ,\nu_{nk}\right)}^{⊤}$, $Y={\left(Y_{1}^{⊤},\ldots ,Y_{n}^{⊤}\right)}^{⊤}$, $c={\left(c_{11},c_{21},\ldots ,c_{nn}\right)}^{⊤}$, $d={\left(d_{1},d_{2}\right)}^{⊤}$, $w_{2}={\left(w_{2,1},\ldots ,w_{2,Q}\right)}^{⊤}$, Σ be an $n^{2}\times n^{2}$ matrix with $b_{ikjl}$ as the (i+(k−1)n,j+(l−1)n) element, and ***T*** be a $n^{2}\times 2$ matrix with $a_{ikj}$ as the (i+(k−1)n,j) element. Then, the PLS (Equation 11) reduces to
(12)$$\begin{array}{c} \frac{1}{n}{∥Y-Td-\Sigma c∥}^{2}+\lambda_{1}\sum_{k=1}^{2}w_{1k}\left|d_{k}\right|+\tau_{0}c^{⊤}\Sigma c+\tau_{1}w_{2}^{⊤}\theta , \end{array}$$
subject to $\theta_{v}\geq 0,\;v=1,2,\ldots ,Q$.

The backfitting algorithm in Wang et al. [12] is applied to solve the PLS (Equation 12) as follows (Algorithm 1):
**Algorithm 1** Model Selection Algorithm Set initial value $d=d^{0}$, $\theta =\theta^{0}$. **repeat**
    Update ***c*** by minimizing $\frac{1}{n}{∥Y-Td-\Sigma c∥}^{2}+\tau_{0}c^{⊤}\Sigma c$    Calculate $Y^{*}=Y-R\theta$, where ***R*** is an n×Q matrix with the *v*-th column being $w_{2,v}^{-1}\Sigma_{v}c$    Update ***d*** by minimizing $\frac{1}{n}{∥Y^{*}-Td∥}^{2}+\lambda_{1}\sum_{k=1}^{2}w_{1k}\left|d_{k}\right|$
    select tuning parameter *M* by the *k*-fold cross-validation or BIC method
    Update θ by minimizing $\frac{1}{n}{∥Y-Td-R\theta ∥}^{2}+\tau_{0}c^{⊤}R\theta$ subject to $\theta_{v}\geq 0$ for 1≤v≤Q and $w_{2}^{⊤}\theta \leq M$
 **until**
***c***, ***d*** and θ converge
 **return**
***c***, ***d*** and θ

## 4. Statistical Properties

In this section, we assume that X and Y are complete measurable spaces. We let *P* be a probability measure on $X^{q}\times L^{2}\left(T\right)$ and $M=T\times X^{q}$. Without the loss of generality, we let the terms $\lambda_{1}\sum_{k=1}^{2}w_{1k}\left|d_{k}\right|$ and $\lambda_{2}\sum_{v=1}^{Q}w_{2,v}{∥P_{v}f∥}_{H}$ in (Equation 6) be combined into ${∥f∥}_{H}$.

We define a loss function,
$$\begin{array}{c} L\left(f;x,y\right)=\int_{0}^{1}{\left(y\left(t\right)-f\left(t,x\left(t,\;\cdot \;\right)\right)\right)}^{2}dt, \end{array}$$
where y(t)∈Y and $x\in X^{q}$. The corresponding L-risk function (Steinwart and Christmann [28]) is
$$\begin{array}{c} R_{L,P}\left(f\right)=E_{P}\left[L\left(f;x,y\right)\right]. \end{array}$$

We let $f^{*}=\underset{f\in H}{\arg\min}R_{L,P}\left(f\right)$, $R_{L,P,H}^{*}=R_{L,P}\left(f^{*}\right)$, and
$$\begin{array}{c} f_{P,\lambda }=\underset{f\in H}{\arg\min}\left\{R_{L,P}\left(f\right)+\lambda {∥f∥}_{H}\right\}. \end{array}$$

Obviously, $\hat{f}=f_{D,\lambda }$. We state convergence properties in the following theorem and show its proofs in Appendix B.

**Theorem** **2.**
*Assume that f:M→R is measurable for any f∈H, M is a complete measurable space, and ${\left|P\right|}_{2}=\int_{X^{q}\times L^{2}\left(T\right)}{∥y(t)∥}_{2}^{2}dP\left(x,y\right)<\infty$. When λ→0 and $\lambda^{6}n\to \infty$ as n→∞, we have*

$$\begin{array}{cc} \; & |R_{L,P}\left(\hat{f}\right)-R_{L,P,H}^{*}|=O_{p}\left(\lambda \right). \end{array}$$



Theorem 2 states that as λ tends to 0 and $\lambda^{6}n$ tends to infinity as *n* tends to infinity, the function estimate $\hat{f}$ is L-risk consistent (Steinwart and Christmann [28]).

## 5. Simulation

In this section, numerical experiments are studied to evaluate the performance of the proposed model selection approach. Functional covariate take $x_{i}\left(t,\;\cdot \;\right)=\left(x_{i1}\left(t,\;\cdot \;\right),x_{i2}\left(t,\;\cdot \;\right)\right)$, where $x_{ij}\left(t,\;\cdot \;\right)=\cos\left(2\pi \left(x_{ij}^{*}\left(t,\;\cdot \;\right)\right)\right)$, and $x_{ij}^{*}\left(t,\;\cdot \;\right)$ follows a Gaussian process with mean function μ(t)=t. Kernel function for the GP takes the RBF kernel $k_{g}\left(s_{1},s_{2}\right)=exp\left(-{\left(s_{1}-s_{2}\right)}^{2}/2\right)$ for j=1 and the rational quadratic kernel $k_{l}\left(s_{1},s_{2}\right)=1/\left(1+{\left(s_{1}-s_{2}\right)}^{2}\right)$ for j=2. Three functions for f(t,x(t,·)) are presented as follows: for t∈ [0, 1],
$$\begin{array}{cc} M_{1}:f\left(t,x\left(t,\;\cdot \;\right)\right)= & 1+5\cos{\left(2\pi t\right)}^{3},\\ M_{2}:f\left(t,x\left(t,\;\cdot \;\right)\right)= & 1+0.5t+10\int_{0}^{1}x_{1}^{3}\left(t,s\right)ds+5\int_{0}^{1}x_{2}^{3}\left(t,s\right)ds+10\int_{0}^{1}x_{1}^{3}\left(t,s\right)ds\int_{0}^{1}x_{2}^{3}\left(t,s\right)ds,\\ M_{3}:f\left(t,x\left(t,\;\cdot \;\right)\right)= & 1+5\cos\left(2\pi t\right)+10\int_{0}^{1}x_{1}\left(t,s\right)x_{2}\left(t,s\right)ds. \end{array}$$
We see that $M_{1}$ has the main effect of *t*, $M_{2}$ consists of three main effects and the nonparametric nonparametric interaction of $x_{1}$ and $x_{2}$, $M_{3}$ consists of the main effect of *t*, and the nonparametric nonparametric interaction of $x_{1}$ and $x_{2}$. Random error $\epsilon_{i}\left(t\right)$ follows $N(0,0.22^{2})$ and $N(0,0.5^{2})$. All simulations are repeated 200 times.

We generate *n* samples $\left\{y_{i}\left(t\right),x_{i}\left(t,\;\cdot \;\right):i=1,\ldots ,n\right\}$ as training data, and $n_{t}=50$ samples $\left\{{\tilde{y}}_{i}\left(t\right),{\tilde{x}}_{i}\left(t,\;\cdot \;\right):i=1,\ldots ,n\right\}$ as test data. For comparison, we evaluate the performance using the following root mean square error (RMSE) on the test data:$$\begin{array}{c} \mathrm{RMSE}=\sqrt{\frac{1}{n_{t}}\sum_{i=1}^{n_{t}}{∥f\left(t,{\tilde{x}}_{i}\left(t,\;\cdot \;\right)\right)-\hat{f}\left(t,{\tilde{x}}_{i}\left(t,\;\cdot \;\right)\right)∥}_{2}^{2}}, \end{array}$$
where ${∥\;\cdot \;∥}_{2}$ is the norm of $L^{2}\left(T\right)$.

The proposed model selection method is used to train the model and predict the test data, denoted by $L_{1}$. Not considering model selection, we use the $L_{2}$ penalty method to estimate the NCRM model denoted by $L_{2}$, which minimizes the following objective function,
(13)$$\begin{array}{c} \frac{1}{n}\sum_{i=1}^{n}\left\{\int_{0}^{1}{\left(y_{i}\left(t\right)-f\left(t,x_{i}\left(t,\;\cdot \;\right)\right)\right)}^{2}dt\right\}+\lambda \sum_{v=1}^{Q}{∥P_{v}f∥}_{H}^{2}, \end{array}$$
where λ is the tuning parameter. After model selection, the selected model is estimated with the $L_{2}$ penalty method, which is denoted by $L_{1}+L_{2}$. Table 1 shows the average RMSE and standard deviation in parentheses for these three estimation methods, $L_{1}$, $L_{2}$ and $L_{1}+L_{2}$. We see that under models M1 and M3, $L_{1}+L_{2}$ have the smallest RMSEs among these three estimation methods. Under model M2, $L_{1}$ has better performance than $L_{2}$ and comparable results with those of $L_{1}+L_{2}$. In addition, for the three different methods, prediction performance improves as the σ decreases and the training sample size increases.

To evaluate the performance of model selection by the $L_{1}$ penalty method, we take three measurement indices in Wang et al. [12], specificity (SPE), sensitivity (SEN) and $F_{1}$ scores,
$$\begin{array}{c} \mathrm{SPE}=\frac{\mathrm{TN}}{\mathrm{TN}+\mathrm{FP}},\;\;\;\mathrm{SEN}=\frac{\mathrm{TP}}{\mathrm{TP}+\mathrm{FN}},\;\;\;F_{1}=\frac{2\mathrm{TP}}{2\mathrm{TP}+\mathrm{FN}+\mathrm{FP}}, \end{array}$$
where TP, TN, FP and FN are the numbers of true positives, true negatives, false positives and false negatives, respectively. The non-zero components of the decomposition of the regression function are considered as positive samples. For $\theta_{v}>0$, its estimated value is larger than 0, which is considered a true positive.

Table 2 shows the sensitivities, specificities, and F1 scores. Overall, the $L_{1}$ penalty method performs well in different simulation settings. In addition, model selection becomes better with decreasing σ and increasing training sample size.

## 6. Application

The proposed model selection approach is applied to analyze stroke rehabilitation data with 70 stroke survivors ([7]).

The data consist of 34 acute patients with an incidence of stroke less than a month ago and 36 chronic patients with an incidence of stroke more than six months ago. To improve upper limb functions for stroke patients, a convenient home-based rehabilitation system via action video games with 3D-position movement behaviors has been developed [7,8]. The patients played the movement game at scheduled times. For each visit time *t*, the impairment level of subject *i* was assessed using a measure called CAHAI (Chedoke Arm and Hand Activity Inventory) score, denoted as $y_{i}\left(t\right)$, and movements such as forward circular movement and sawing movement were recorded. In this paper, three movements, forward circular movement of the parental limb from the *x*-axis ($x_{i1}=LA05.lx$), sawing movement of the parental limb from the *y*-axis ($x_{i2}=LA09.ly$), and the movement of the non-parental limb from the direction of the *x*-axis ($x_{i3}=LA28.rqx$) are taken as functional covariates. For the purpose of illustrating the proposed method, we use the data from acute patients. During the three-month study period, each acute patient received up to seven assessments, which resulted in 173 observations. CAHAI scores were normalized before analysis.

In this paper, we focus on the interaction effect upon the order of two, and take the following decomposition: $$\begin{array}{cc} K_{0}\left(\left(t,u\right),\left(t^{\prime },u^{\prime }\right)\right) & =1+k_{1}\left(t\right)k_{1}\left(t^{\prime }\right),\\ K_{1}\left(\left(t,u\right),\left(t^{\prime },u^{\prime }\right)\right) & =K_{2}^{\left(1\right)}\left(t,t^{\prime }\right),\\ K_{1+j}\left(\left(t,u\right),\left(t^{\prime },u^{\prime }\right)\right) & =K_{2,j}^{\left(2\right)}\left(u_{j},u_{j}^{\prime }\right),\\ K_{4+j}\left(\left(t,u\right),\left(t^{\prime },u^{\prime }\right)\right) & =k_{1}\left(t\right)k_{1}\left(t^{\prime }\right)K_{2,j}^{\left(2\right)}\left(u_{j},u_{j}^{\prime }\right)+K_{2}^{\left(1\right)}\left(t,t^{\prime }\right)K_{2,j}^{\left(2\right)}\left(u_{j},u_{j}^{\prime }\right),\\ K_{5+j+l}\left(\left(t,u\right),\left(t^{\prime },u^{\prime }\right)\right) & =K_{2,j}^{\left(2\right)}\left(u_{j},u_{j}^{\prime }\right)K_{2,l}^{\left(2\right)}\left(u_{l},u_{l}^{\prime }\right), \end{array}$$
for j,l=1,2,3 and j<l. Readers can also choose various kinds of SS ANOVA decomposition by merging kernel functions according to their own needs. From Section 3, we have
$$K^{*}\left(\left(t,u\right),\left(t^{\prime },u^{\prime }\right)\right)=\sum_{v=1}^{10}w_{2,v}^{-1}\theta_{v}K_{v}\left(\left(t,u\right),\left(t^{\prime },u^{\prime }\right)\right),$$
where coefficient $\theta_{v}$ for kernel function $K_{v}$ provides levels of contribution of $K_{v}$ to the overall model.

The penalty method with $L_{1}$ regularization for model selection is applied to stroke rehabilitation data. Parameters $\left\{\theta_{v}\right\}$ are computed, and values larger than 0 are $\theta_{2}=4.157$, $\theta_{3}=0.819$, $\theta_{4}=0.636$, $\theta_{7}=0.592$ and $\theta_{10}=0.741$. This shows that the main effects of $x_{i1}\left(t,\;\cdot \;\right)$, $x_{i2}\left(t,\;\cdot \;\right)$ and $x_{i3}\left(t,\;\cdot \;\right)$, the linear nonparametric interaction of *t* and $x_{3}\left(t,\;\cdot \;\right)$ and the nonparametric-nonparametric interaction of $x_{2}\left(t,\;\cdot \;\right)$ and $x_{3}\left(t,\;\cdot \;\right)$ have nonzero contributions to the CAHAI score. Thus, the three movements, forward circular movements of the parental limb, awing movements of the parental limb and of the non-parental limb, may be helpful to the recovery of stroke patients. In addition, the interaction of awing movements of the parental limb and the non-parental limb, may contribute to the level of daily life dependence or upper limb function impairment. Figure 1 plots the estimates of nonparametric regression functions for four stroke patients. We can see that the regression function in the NCRM model has the same trend as the scores of CAHAI, and on the whole, they all show an increasing trend along with fluctuating trends, which shows that movements may improve upper limb function for stroke patients.

Prediction performance of the proposed method is evaluated using a tenfold cross-validation,
$$\begin{array}{c} \mathrm{RPE}=\frac{1}{10}\sum_{i=1}^{10}\sqrt{\frac{1}{n_{j}}\sum_{i\in jth\;fold}^{\;}{∥Y_{i}\left(t\right)-{\hat{Y}}_{i}^{\left(-j\right)}\left(t\right)∥}_{2}^{2}}, \end{array}$$
where ${\hat{Y}}_{i}^{\left(-j\right)}\left(t\right)$ is the predicted value of $Y_{i}\left(t\right)$ based on the fitted selected model with the $L_{1}+L_{2}$ penalty and the data excluding the *j*th fold. The RPE for Stoke data is 1.0690, which is smaller than 1.1700 from the method of Zhai et al. [9].

## 7. Conclusions

For functional data with functional covariate inputs, this paper applies the Gaussian kernel function to construct the tensor product RKHS to model the regression function. This leads to a nonparametric concurrent regression model. The $L_{1}$ penalty method is used to detect components of the SS ANOVA decomposition of the regression function, which has nonzero contribution to model fit. The backfitting algorithm is developed to estimate the model selection. The proposed method is applied to stroke rehabilitation data, and the results show that besides the main effects, there are interaction effects of the movements on the CAHAI score. This indicates that movements may help improve the level of daily life dependence or impairment of upper limb function of a stroke patient.

## Figures and Tables

**Figure 1 entropy-25-01327-f001:**
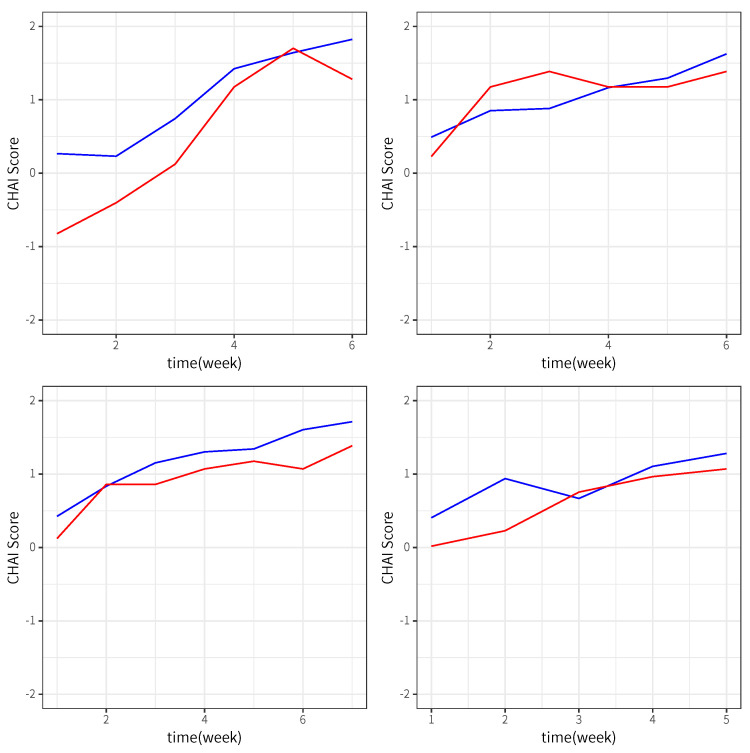
CAHAI scores (red line) and their corresponding fitted values (blue line) for 4 patients.

**Table 1 entropy-25-01327-t001:** Average values and standard deviations of RMSEs. (Methods corresponding to bold numbers perform best).

*n*	σ	Model	$L_{1}$	$L_{2}$	$L_{1}+L_{2}$
20	0.2	M1	0.2906 (0.1386)	0.3010 (0.0287)	0.0981(0.0420)
		M2	0.8876(0.1147)	0.9577 (0.0782)	0.9493 (0.0707)
		M3	1.2065 (0.8556)	0.9423 (0.1713)	0.7108(0.6317)
	0.5	M1	0.3027 (0.1323)	0.4238 (0.0654)	0.1205(0.0591)
		M2	0.9486(0.1166)	1.0039 (0.0760)	0.9996 (0.0794)
		M3	1.3906 (0.9134)	1.0044 (0.1952)	0.8008(0.5486)
40	0.2	M1	0.1655 (0.0127)	0.2390 (0.0416)	0.0792(0.0113)
		M2	0.7722(0.0517)	0.8088 (0.0877)	0.7928 (0.0630)
		M3	0.6664 (0.2255)	0.5773 (0.0456)	0.3968(0.0423)
	0.5	M1	0.1913 (0.0944)	0.3257 (0.0860)	0.0913(0.0179)
		M2	0.8605(0.0678)	0.8918 (0.0856)	0.8819 (0.0746)
		M3	0.7824 (0.2060)	0.6869 (0.0420)	0.4952(0.0764)
80	0.2	M1	0.1358 (0.0860)	0.1416 (0.0238)	0.0737(0.0076)
		M2	0.6040(0.0620)	0.6599 (0.0894)	0.6595 (0.0708)
		M3	0.3227 (0.0218)	0.3793 (0.0237)	0.2635(0.0248)
	0.5	M1	0.1635 (0.1412)	0.2419 (0.0704)	0.0844(0.0206)
		M2	0.7338(0.0552)	0.7589 (0.0696)	0.7578 (0.0605)
		M3	0.4080 (0.1488)	0.5511 (0.0475)	0.3566(0.0470)

**Table 2 entropy-25-01327-t002:** Average values and standard deviations of SPE, SEN, F1 scores under models M1, M2, M3.

*n*	σ	Model	SPE	SEN	F1
20	0.2	M1	0.9956 (0.0291)	0.9800 (0.1404)	0.9783 (0.1421)
		M2	0.9700 (0.0601)	0.7017 (0.1472)	0.7883 (0.1166)
		M3	0.9906 (0.0366)	0.9850 (0.1219)	0.9592 (0.1527)
	0.5	M1	0.9961 (0.0233)	0.9850 (0.1219)	0.9800 (0.1279)
		M2	0.9786 (0.0511)	0.6917 (0.1529)	0.7881 (0.1201)
		M3	0.9889 (0.0386)	0.9850 (0.1219)	0.9553 (0.1543)
40	0.2	M1	1.0000 (0.0000)	1.0000 (0.0000)	1.0000 (0.0000)
		M2	0.9614 (0.0667)	0.9817 (0.0762)	0.9512 (0.0872)
		M3	0.9833 (0.0428)	1.0000 (0.0000)	0.9517 (0.1212)
	0.5	M1	0.9989 (0.0111)	0.9950 (0.0707)	0.9933 (0.0744)
		M2	0.9407 (0.0720)	0.9600 (0.1086)	0.9177 (0.1060)
		M3	0.9844 (0.0417)	1.0000 (0.0000)	0.9550 (0.1178)
80	0.2	M1	0.9983 (0.0175)	1.0000 (0.0000)	0.9958 (0.0424)
		M2	0.9743 (0.0737)	1.0000 (0.0000)	0.9635 (0.0701)
		M3	1.0000 (0.0000)	1.0000 (0.0000)	1.0000 (0.0000)
	0.5	M1	0.9956 (0.0291)	0.9850 (0.1219)	0.9817 (0.1259)
		M2	0.9546 (0.0853)	1.0000 (0.0000)	0.9317 (0.1031)
		M3	0.9983 (0.0135)	1.0000 (0.0000)	0.9950 (0.0406)

## Data Availability

The data presented in this study are available on reasonable request from the corresponding author.

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
