# Peer review of "Detection of Interaction Effects in a Nonparametric Concurrent Regression Model"

_entropy, 2023, doi:10.3390/e25091327_

Round 1

Reviewer 1 Report

A lot of work exists regarding smoothing spline anova, reproducing kernel Hilbert spaces and l1 regularization.
Especially the work by Grace Wahba is well-known in this domain. Surprisingly, not a single reference to her work is present in this paper.
In one work already from 2004, Journal of the American statistical Association, smoothing spline ANOVA, RKHS and lasso (l1 regularization) were introduced and studied.
"Variable Selection and Model Building via Likelihood Basis Pursuit", Zhang, H., Wahba, G., Lin, Y., Voelker, M., Ferris, M., Klein, R. and Klein, B., JASA 99 (2004) 659-672.
In that 2004 paper the interactions explicitly appeared.
Several other relevant papers are listed on Wahba's website with publications.

The current submission heavily leans upon Wang, Zhanfeng; Dong, H.; Ma, P.; Wang, Y. (2022), JGCS. While a lot of formulas appear in this paper, many are elaborated versions of model equations in the form of the 2004 paper of Zhang et al., and as such not difficult. The presence of interactions makes them long. No theoretical results are derived here, a reader is referred to the 2022 paper.

The abstract states "We then develop a model selection method [...]". In the paper this 'development' boils down to adding an l1 regularization, something which is already introduced in the smoothing spline models in the 2004 paper by Zhang et al. and hence since appeared in many different models. While this specific model might not have been described, there is no new 'development' in this paper, no new model selection method is developed.

This is an application and combination of existing results (some about 20 years old), some more recent. It might be nice for the people working with those data that this method provides an analysis, but this should not be seen as a more methodological contribution since the level of novelty is very low.

-

Author Response

 We are very grateful to the reviewer for careful reviews and constructive comments. The attached are our responses to each of these comments. We have highlighted most of the revisions by blue colour in the revised manuscript. 

Reviewer 2 Report

I had only a general remark written in the report file. 

Author Response

(The authors gave the same response as above.)

Reviewer 3 Report

Pan et al. have proposed a nonparametric concurrent regression model to detect interaction effects among functional covariates. Detection of the main and interaction effects have been performed through regularization on the terms from the SS ANOVA decomposition of the tensor product space.  

As shown in Section 2, the decomposition to the different types of main and interaction effects is complicated, including the linear-nonparametric and nonparametric-nonparametric interactions. Are these decomposition necessary and supported by the data? Usually, in nonlinear interaction studies, exploratory data analysis can quickly justify the necessity of the interaction effects (See Ren et al. 2020). In addition, choosing order of the decomposition appears arbitrary. In the case study, why not choose order 3 or even higher order? Is there a model selection procedure to determine the optimal order?

The authors overlooked Bayesian analysis on interaction studies. For example, in Ren et al. 2020, the statistical inference can be conducted to see whether the nonlinear interaction is indeed significant by constructing confidence intervals. The regularized estimation results in Section 5 have not been provided with significance measures so it’s not clear whether they differ from 0 significantly. In Figure 1, are there any confidence intervals generated for the curves?

What’s the dimension subject to selection when q=3 in section 2? In the case study, the dimension subject to selection is 10. Please report the computation time. There are no alternative methods applied in the case study so it’s not clear if the proposed NCRM has the best performance.  

Are the L1, L2 and L1+2 all proposed by this study? What are the alternative methods?

Ren, J., Zhou, F., Li, X., Chen, Q., Zhang, H., Ma, S.,  & Wu, C. (2020). Semiparametric Bayesian variable selection for gene‐environment interactions. Statistics in medicine, 39(5), 617-638.

Minor editing of English language required.

Author Response

(The authors gave the same response as above.)

Reviewer 4 Report

please see report

please see report

Author Response

(The authors gave the same response as above.)

Round 2

Reviewer 1 Report

Some text has been added, but the scientific content stayed the same. The paper has only a very small incremental contribution to the literature.

/

Author Response

In this revision, we have revised introduction section to more clearly state our contributions, and have studied statistical properties of the function estimate, such as L-risk consistence. 

Reviewer 3 Report

Thanks for the revision. No further comments.

Minor editing of English language required.

Author Response

Thanks, we have revised the manuscript carefully. 

Reviewer 4 Report

This revised version has much improvement in clearly identifying the contributions and responding to the comments of this referee.  This revised version can be recommended for acceptance.

Author Response

Thanks.